# L-Serine-Modified Poly-L-Lysine as a Biodegradable Kidney-Targeted Drug Carrier for the Efficient Radionuclide Therapy of Renal Cell Carcinoma

**DOI:** 10.3390/pharmaceutics14091946

**Published:** 2022-09-14

**Authors:** Hidemasa Katsumi, Sho Kitada, Shintaro Yasuoka, Rie Takashima, Tomoki Imanishi, Rina Tanaka, Satoru Matsuura, Hiroyuki Kimura, Hidekazu Kawashima, Masaki Morishita, Akira Yamamoto

**Affiliations:** 1Department of Biopharmaceutics, Division of Clinical Pharmaceutical Sciences, Kyoto Pharmaceutical University, Yamashina-ku, Kyoto 607-8414, Japan; 2Department of Analytical and Bioinorganic Chemistry, Division of Analytical and Physical Sciences, Kyoto Pharmaceutical University, Yamashina-ku, Kyoto 607-8414, Japan; 3Radioisotope Research Center, Kyoto Pharmaceutical University, Yamashina-ku, Kyoto 607-8414, Japan

**Keywords:** kidney-targeted delivery, radionuclide therapy, renal cell carcinoma, L-serine, yttrium (^90^Y)

## Abstract

In the present study, L-serine (Ser)-modified poly-L-lysine (PLL) was synthesized to develop a biodegradable, kidney-targeted drug carrier for efficient radionuclide therapy in renal cell carcinoma (RCC). Ser-PLL was labeled with ^111^In/^90^Y via diethylenetriaminepentaacetic acid (DTPA) chelation for biodistribution analysis/radionuclide therapy. In mice, approximately 91% of the total dose accumulated in the kidney 3 h after intravenous injection of ^111^In-labeled Ser-PLL. Single-photon emission computed tomography/computed tomography (SPECT/CT) imaging showed that ^111^In-labeled Ser-PLL accumulated in the renal cortex following intravenous injection. An intrarenal distribution study showed that fluorescein isothiocyanate (FITC)-labeled Ser-PLL accumulated mainly in the renal proximal tubules. This pattern was associated with RCC pathogenesis. Moreover, ^111^In-labeled Ser-PLL rapidly degraded and was eluted along with the low-molecular-weight fractions of the renal homogenate in gel filtration chromatography. Continuous Ser-PLL administration over five days had no significant effect on plasma creatinine, blood urea nitrogen (BUN), or renal histology. In a murine RCC model, kidney tumor growth was significantly inhibited by the administration of the beta-emitter ^90^Y combined with Ser-PLL. The foregoing results indicate that Ser-PLL is promising as a biodegradable drug carrier for kidney-targeted drug delivery and efficient radionuclide therapy in RCC.

## 1. Introduction

Renal cell carcinoma (RCC) is a kidney cancer originating in the lining of the proximal tubules and is reputed to be the most lethal malignant urological tumor [1]. The current standard treatments for RCC include inhibitors of tyrosine kinase, mTOR, and immune checkpoints. As these therapies have several pharmacokinetic issues, however, their efficacy and safety are questionable [2,3,4]. It is therefore necessary to develop a drug delivery system (DDS) for the selective transport of therapeutic agents, particularly to the proximal tubule associated with RCC pathogenesis. Nevertheless, it is difficult to develop a drug carrier with high renal selectivity simply by using the conventional kidney-targeted approach of DDS technology. Furthermore, there are few reports of the successful development of renal-targeted DDS with superior efficacy and safety [5,6,7].

Recently, we observed that after L-serine (Ser)-modified polyamideamine (PAMAM) dendrimer (Ser-PAMAM) was intravenously injected into mice, it accumulated in the proximal tubules of the renal cortex. In Ser-PAMAM, multiple Ser moieties are covalently bound to the PAMAM surface [8,9,10]. Ser is a biological component and its physicochemical properties are well known. Thus, it is considered superior to conventional kidney-targeted moieties in terms of safety and ease of synthesis. PAMAM is familiar as a novel artificial dendritic macromolecular drug carrier and is anticipated for clinical use [11]. It is unknown, however, whether Ser modification could be applied in a kidney-targeted delivery system using linear macromolecular drug carriers. In addition, little is known about the biocompatibility and biodegradability of the inner core of the drug carrier in this kidney-targeted drug delivery system.

In the present study, we selected linear poly-L-lysine hydrobromide (PLL) as a carrier backbone and modified it with Ser to obtain Ser-modified poly-L-lysine (Ser-PLL). Ser-PLL consists almost exclusively of amino acids. Hence, it is expected to have high biocompatibility and biodegradability.

The indium (^111^In) (imaging)/yttrium (^90^Y) (therapy) combination has been evaluated in cancer radiotheranostics [12,13,14]. Recently, ^111^In/^90^Y-loaded ibritumomab (anti-CD20 antibody) thiuxetane was approval by the United States Food and Drug Administration and has been used to treat CD20-positive relapsed or refractory low-grade B-cell non-Hodgkin’s lymphoma and mantle cell lymphoma. In this formulation, ibritumomab is bound to ^111^In/^90^Y via the chelating agent (1-(2)-methyl-4-isocyanatobenzyl-diethylenetriamine-N,N,N′,N″,N″-pentaacetic acid (MX-DTPA; thiuxetane) [15,16]. We hypothesized that an imaging/therapy combination involving ^111^In/^90^Y may facilitate the development of an innovative RCC treatment modality.

The aims of this study were to develop a biodegradable kidney-targeted drug carrier using the amino acids Ser and Lys and evaluate its efficacy as a radionuclide therapy for RCC. We endeavored to develop a kidney-targeted radiotheranostics system using Ser-PLL with ^111^In/^90^Y via the chelating agent diethylenetriaminepentaacetic acid (DTPA). We intravenously injected ^111^In-labeled Ser-PLL in mice and analyzed its tissue distribution. We then assessed the efficacy of ^90^Y-labeled Ser-PLL at preventing increases in the number of tumor cells in the kidneys of a mouse RCC model.

## 2. Materials and Methods

### 2.1. Chemicals and Reagents

Poly-L-lysine hydrobromide (PLL) (MW range = 4000–15,000) and polyamidoamine (PAMAM) dendrimer with an ethylenediamine core (generation (G) 3) in methanol were purchased from Sigma-Aldrich Corp. (St. Louis, MO, USA). Boc-Ser(tBu)-OH and HOBt were purchased from Watanabe Chemical Industries (Hiroshima, Japan). Hexafluorophosphate benzotriazole tetramethyl uronium (HBTU) was purchased from Merck Millipore (Burlington, MA, USA). Dimethyl sulfoxide (DMSO; super dehydrated grade), N,N-dimethylformamide (DMF; super dehydrated grade), N,N-diisopropylethylamine (DIPEA), piperidine, trifluoroacetic acid (TFA), triisopropylsilane (TIS), Turk’s solution, and poly-L-lysine hydrobromide (MW range = 15,000–30,000) were purchased from Fujifilm Wako Pure Chemical Industries Ltd. (Osaka, Japan). PD-10 was purchased from GE Healthcare Japan (Tokyo, Japan). The ^111^InCl_3_ was supplied by Nihon Medi-Physics (Tokyo, Japan). ^90^YCl_3_ was purchased from Eckert Radiopharma (Berlin, Germany). DTPA anhydride was purchased from Chemical Dojin Co., Ltd. (Kumamoto, Japan). All other chemicals were of commercial reagent grade.

### 2.2. Animals

Male ddY mice (age 5 wks; average weight 25 g) and male Balb/c mice (age 5 wks; average weight 20 g) were purchased from Japan SLC (Shizuoka, Japan). The animals were maintained under conventional housing conditions. All animal experiments were conducted according to the principles and procedures outlined in the National Institutes of Health Guide for the Care and Use of Laboratory Animals. The Animal Experimentation Committee of the Kyoto Pharmaceutical University approved this experimental protocol.

### 2.3. Synthesis of Ser-Poly-L-Lysine

Ser-PLL was synthesized by reacting Ser with PLL (MW range = 4000–15,000) using the previously published HBTU-HOBt method with slight modifications [8,17] (Appendix A). Briefly, PLL (MW range = 4000–15,000) was coupled to its surface amino groups with 1.1 eq (equivalent to the surface amino group of PLL) Boc-Ser(t-Bu)-OH in DMF/DMSO (3:1) by mixing it with 1.1 eq HBTU, 1.1 eq HOBt, and DIPEA to adjust the pH to 8−9. The mixture was stirred at room temperature until a ninhydrin test yielded negative results for thin-layer chromatography (TLC) analysis. After the coupling, the reaction mixture was evaporated and the product was dissolved in chloroform. The organic phase was washed three times with 5% NaHCO_3_ and saturated with sodium chloride thrice. Then, the organic phase was dried over anhydrous MgSO_4_ and vacuum-filtered and evaporated. The precipitates were then dissolved in a trifluoroacetic acid (TFA) cocktail (95% (*v*/*v*) TFA, 2.5% (*v*/*v*) thioanisole (TIS), and 2.5% (*v*/*v*) purified water) to deprotect the Boc and *t*-Bu groups and protect the Ser oxygen atom. The mixture was then incubated at room temperature for 90 min. After deprotection and evaporation, the crude precipitate was dispersed in ultrapure water. After dialysis with ultrapure water overnight, the solution was lyophilized to obtain Ser-PLL, which was identified using a ^1^H nuclear magnetic resonance (NMR) spectroscopy (AM-300 FT-NMR spectrometer; Bruker Corp., Billerica, MA, USA) in deuterated water (D_2_O). In the PLL ^1^H NMR spectrum, peaks were observed at δ 4.18 (s, α–CH), 2.81 (m, ε–CH_2_), 1.82–1.05 (s, β–CH_2_, γ–CH_2_, and δ–CH_2_) [18]. In the Ser-PLL ^1^H NMR spectrum, the peaks corresponding to Ser appeared at δ 3.72–3.85 (m, CH_2_), and the integral ratio of Ser methylene protons to PLL methylene protons indicated that the desired product had been obtained (the degree of modification of the Ser moiety linked to the amino groups of PLL was approximately 97%) (Appendix A). As the polymerization degree of PLL is estimated to be approximately 19–72 units with a MW range of 4000–15,000, these results indicate that the average composition of Ser units of the obtained Ser-PLL was 45, with an approximate Ser-PLL molar mass of 10 kDa.

Ser-PLL was then dissolved in phosphate-buffered saline (PBS) (pH 7.4), and its mean particle diameter and zeta potential were measured with a Zetasizer PRO Blue Label (Malvern Instruments, Worcestershire, UK). To synthesize high-MW Ser-PLL (Ser-PLL (H)), we reacted PLL (MW = 15,000–30,000) with Boc-Ser(t-Bu)-OH using the preceding methods. Ser-modified polyamidoamine dendrimer (G3) (Ser-PAMAM conjugating ~32 Ser molecules) was synthesized and used as a control according to the previously reported HBTU-HOBt method [8].

### 2.4. Tissue Distribution of ^111^In-Labeled Ser-Poly-L-Lysine

Ser-PLL was labeled with ^111^In using DTPA according to a previously published method to determine the tissue distribution of Ser-PLL [19,20]. The ^111^In-labeled Ser-PLL was intravenously injected at 1 mg/kg dose into the tail vein of each ddY mouse. At appropriate time points after the injection, blood was collected from the abdominal vena cava under isoflurane anesthesia. The liver, kidneys, spleen, heart, and lungs were excised, rinsed with saline, blotted dry, and weighed. The blood was centrifuged at 2000× *g* for 5 min to obtain the plasma. The organ samples and 100 μL plasma were transferred to counter tubes and their radioactivity levels were measured with a gamma counter (1480 WizardTM 3′; PerkinElmer, Boston, MA, USA). The tissue distribution of PLL was determined as previously described, using ^111^In-labeled PLL as a control.

### 2.5. In Vivo SPECT/CT Imaging of ^111^In-Labeled Ser-Poly-L-Lysine Tissue Distribution

Single-photon emission computed tomography/computed tomography (SPECT/CT) was performed with a NanoSPECT/CT (Bioscan Inc., Washington, DC, USA) according to a previously reported method [10]. Briefly, ^111^In-labeled Ser-PLL (15.6 MBq/mouse) was intravenously injected into a ddY mouse. A CT scan of the mouse was performed under isoflurane anesthesia for anatomical reference. Three hours after the injection, a 1-h SPECT scan was obtained under isoflurane anesthesia. The SPECT image was reconstructed and analyzed with VivoQuant v. 5.1 (inviCRO, Hillsboro, OR, USA).

### 2.6. Intrarenal Distribution of FITC-Labeled Ser-Poly-L-Lysine

Ser-PLL was labeled with fluorescein isothiocyanate (FITC), as previously reported [8]. The FITC-labeled Ser-PLL was then injected into the tail vein of each ddY mouse. After 60 min, the kidneys were excised under isoflurane anesthesia, immersed with 30% sucrose solution. After the immersion at 4 °C overnight, the kidneys were fixed with optimal cutting temperature (OCT) compound, and frozen. The frozen kidney sections were stained with 10 μg/mL of 4′,6-diamidino-2-phenylindole (DAPI) (Fujufilm Wako Pure Chemical Industries Ltd.). The stained kidney sections were observed under a laser-scanning confocal microscope (Nikon A1R, Nikon Corp.; Tokyo, Japan).

### 2.7. Biodegradability of Ser-Poly-L-Lysine

Ser-PLL biodegradability was evaluated using a previously published method, with slight modifications [21]. Briefly, ^111^In-labeled Ser-PLL, ^111^In-labeled PLL (control), ^111^In-labeled PAMAM (G3) (control), and ^111^In-labeled Ser-PAMAM (G3) (control) were intravenously injected at 1 mg/kg dose into the tail vein of each ddY mouse and the animals were sacrificed 3 h later by abdominal vena cava amputation under isoflurane anesthesia. Residual blood in each kidney was removed by passing saline through the left ventricle. Each kidney was then immediately ice-cooled, combined with 4 mL purified water, and homogenized. Saturated KCl solution (1 mL) was added to each kidney homogenate. The suspension was allowed to stand at 4 °C overnight and then centrifuged. The supernatant (1 mL) was eluted with a PD-10 column and 0.1 M acetate buffer (pH 6.0). The radioactivity levels of each fraction (10 drops, 0.35 mL) were measured. The high- and low-MW fractions were determined based on the elution patterns.

### 2.8. Ser-Poly-L-Lysine Nephrotoxicity in Mice

Ser-PLL (1 mg/kg/d) was continuously injected intravenously into the tail vein of each ddY mouse for 5 d and acute nephrotoxicity was evaluated [8]. Six days after the first intravenous injection, blood was collected from the vena cava, the kidneys were isolated under isoflurane anesthesia, and the mice were sacrificed. Plasma creatinine was measured with a commercially available kit (Lab Assay; Fujifilm Wako Pure Chemical Industries Ltd.). Blood urea nitrogen (BUN) was measured with a commercially available kit (DIUR-100; BioAssay Systems, Hayward, CA, USA). The kidneys were then excised, fixed with 4% (*v*/*v*) buffered paraformaldehyde (PFA), and embedded in paraffin blocks. Then 5-µm sections were cut with a microtome from the paraffin blocks. The kidney sections were stained with hematoxylin and eosin (H&E), and hepatic injury was evaluated under a light microscope (Biozero; KEYENCE, Osaka, Japan). HgCl_2_ was subcutaneously injected at a dose of 8 mg/kg, and the mouse group bred for 2 d served as a positive control.

### 2.9. Effects of ^90^Y-Labeled Ser-Poly-L-Lysine on Tumor Growth in a Mouse Renal Cell Carcinoma Model

Ser-PLL was labeled with ^90^Y using DTPA and the method applied for ^111^In-labeling [14,19]. The RCC model was established using a previously published method with slight modifications [22]. Briefly, the RCC model was produced by injecting 2.5 × 10^5^ firefly luciferase gene-labeled Colon/26 (Colon26/Luc) cells along with Matrigel^®^ matrix (Corning, NY, USA) into the right renal cortex of each Balb/c mouse under isoflurane anesthesia. Then PBS, free ^90^Y and ^90^Y-labeled Ser-PLL was immediately injected into the tail vein at a dose of 0.3 MBq/mouse. After 14 d, blood was collected from each vena cava and the kidneys were isolated under isoflurane anesthesia. The right kidney was homogenized in lysis buffer and its luciferase activity was measured with a luminometer (Lumat LB9507; EG&G Berthold, Württemberg, Germany). The cancer cells in each kidney were enumerated based on the luciferase activity and by using a regression line, as previously reported [23,24].

Blood was transferred to a microtube containing ethylenediamine-N,N,N′,N′-tetraacetic acid dipotassium salt dihydrate (EDTA-2K) and diluted tenfold with Turk’s solution (Fujifilm Wako Pure Chemical Industries Ltd.). The white blood cells were then enumerated on a counting plate (Improved NEUBAUER; Elma Sales Co., Ltd., Saitama, Japan) under a microscope [25]. Sections of the untreated left kidneys were examined, and plasma creatinine was measured according to previously described methods.

### 2.10. Statistical Analysis

Statistical significance was analyzed using Student’s *t*-test for two independent groups at a significance level of *p* < 0.05 and Dunnett’s test for multiple comparisons with *p* < 0.05 as the minimum level of significance.

## 3. Results

### 3.1. Physicochemical Properties of Ser-Poly-L-Lysine

Table 1 shows the physicochemical properties of Ser-PLL and PLL. The mean particle diameters of PLL and Ser-PLL were 3.3 ± 0.5 and 4.1 ± 0.9 nm, respectively. The zeta potentials of PLL and Ser-PLL were 8.9 ± 1.6 and 6.6 ± 3.7 mV, respectively.

### 3.2. Tissue Distribution of ^111^In-Labeled Ser-Poly-L-Lysine

Figure 1 shows the tissue distribution of ^111^In-labeled Ser-PLL following intravenous injection. Unmodified ^111^In-labeled PLL quickly disappeared from the plasma and was distributed to the liver and kidneys to levels of ~39% and ~23% of the dose, respectively, after 3 h. By contrast, ^111^In-labeled Ser-PLL accumulated mainly in the kidneys and ~91% of the original dose was retained there. Kidney accumulation was inversely proportional to the MW of Ser-PLL (Figure 1 and Appendix A). Ser-PAMAM accumulated mainly in the kidneys to levels of ~84% after 3 h in a similar manner to our previous study (Appendix A) [8].

### 3.3. Biodistribution Imaging of ^111^In-Labeled Ser-Poly-L-Lysine

Figure 2 shows SPECT/CT images of tissue distribution of ^111^In-labeled Ser-PLL after intravenous injection. The ^111^In-labeled Ser-PLL accumulated mainly in the renal cortex.

### 3.4. Intrarenal Distribution of FITC-Labeled Ser-Poly-L-Lysine

An evaluation of the intrarenal accumulation by FITC labeling clearly displayed fluorescence intensity derived from FITC-labeled Ser-PLL in the renal cortex (Figure 3A) but none in the renal medulla (data not shown). Enlarged microscopic images of the renal cortex exhibited high fluorescence intensity derived from FITC-labeled Ser-PLL in the proximal tubules (Figure 3B).

### 3.5. Ser-Poly-L-Lysine Biodegradability

The biodegradability of ^111^In-labeled Ser-PLL was evaluated using the elution profiles of ^111^In-radioactivity recovered from the kidney homogenates (Figure 4). Relatively high ^111^In-radioactivity was detected in the high-MW fractions after the elution of intact ^111^In-labeled Ser-PLL, PLL, Ser-PAMAM, and PAMAM. The ^111^In-radioactivity partially shifted to the low-MW fraction. Nonetheless, it remained in the high-MW fraction after the elution of the kidney homogenates derived from mice injected with ^111^In-labeled Ser-PAMM and PAMAM (Figure 4A,B). In contrast, the ^111^In-radioactivity shifted to the low-MW fraction after the elution of the kidney homogenates derived from the mice injected with ^111^In-labeled Ser-PLL and PLL (Figure 4C,D).

### 3.6. Ser-Poly-L-Lysine Nephrotoxicity

Figure 5 shows the renal toxicity of Ser-PLL after continuous intravenous injection in mice. HgCl_2_ administration significantly increased the nephrotoxicity indicators plasma creatine and BUN. By contrast, Ser-PLL did not increase the foregoing biomarkers after continuous intravenous injection for 5 d (Figure 5A,B). The kidney sections disclosed severe gap junction damage and necrosis (arrows) after the HgCl_2_ treatment. However, Ser-PLL had negligible adverse effects on renal structure, and the tissue from Ser-PLL-treated mice was similar to that from PBS-treated and naïve mice (Figure 5C).

### 3.7. Therapeutic Potential and Safety of ^90^Y-Labeled Ser-Poly-L-Lysine in Mouse Model of Renal Cell Carcinoma (RCC)

Figure 6 shows the therapeutic potential of ^90^Y-labeled Ser-PLL in a mouse RCC model. Intravenously injected free ^90^Y had a negligible effect on kidney tumor growth. By contrast, intravenously injected ^90^Y-labeled Ser-PLL significantly suppressed any increase in the number of kidney tumor cells.

Intravenously injected free ^90^Y had a negligible effect on the white blood cell counts but significantly increased plasma creatine. In contrast, intravenously injected ^90^Y-labeled Ser-PLL had a negligible effect on plasma creatine or the white blood cell counts (Figure 7A,B). Partial glomerular damage and necrosis (arrow) was observed in the kidney sections of the free ^90^Y group. By contrast, no severe damage was detected in the kidney sections of the ^90^Y-labeled Ser-PLL group (Figure 7C).

## 4. Discussion

In the present study, we successfully developed and evaluated Ser-PLL as a biodegradable kidney-targeted drug carrier to be administered as radionuclide therapy for renal cell carcinoma (RCC). We previously reported that Ser-PAMAM accumulated mainly in the kidneys, and the hydroxyl and amino groups of Ser play important roles in kidney targeting [8]. For this reason, we coupled the amino groups of PLL with the carboxyl groups of Ser to form Ser-PLL, which bears both hydroxyl and amino groups. Ser-PLL had slightly greater kidney accumulation than Ser-PAMAM (Appendix A), possibly because the estimated average number of conjugated Ser residues was 45 in Ser-PLL, whereas Ser-PAMAM conjugated only 32. The number of conjugated Ser residues is proportional to renal affinity [8]. These results indicate that Ser modification could be applied in a kidney-targeted delivery system using linear macromolecular drug carriers in addition to branched macromolecular drug carriers.

Kidney targeting is challenging because most drug nanocarriers are trapped in the hepatic and splenic reticuloendothelial systems. It is essential to maintain the balance between target site affinity and non-target site elimination [26,27]. Here, the hydroxyl groups of Ser might have eluded recognition by the hepatic and splenic reticuloendothelial systems as they are readily hydrated and induce a cage effect against the reticuloendothelial system [28]. In our previous study, we demonstrated that Ser-PAMAM was distributed in the proximal tubules via glomerular filtration. Micropinocytosis and caveola-mediated endocytosis have been implicated in the affinity of Ser-PAMAM for proximal tubules [8]. Therefore, we postulate that Ser-PLL is distributed in the proximal tubules in the same manner as Ser-PAMAM, due to the size of Ser-PLL (~5 nm), which is smaller than the glomerular filtration size cutoff (10 nm) [29,30,31]. Moreover, both Ser-PLL and Ser-PAMAM have the same surface functional groups. The foregoing results of Ser-PLL (PLL MW = 4000–15,000) (Figure 1) and the low renal accumulation of Ser-PLL (H) (PLL MW range = 15,000–30,000) (Appendix A) suggest that Ser-PLL (PLL MW = 4000–15,000) is the optimal size for kidney-targeted delivery mediated by glomerular filtration.

Post-elution ^111^In radioactivity detection by gel filtration chromatography was established as a method to assess the biodegradability of macromolecular drug carriers [21]. Akamatsu et al. (1998) evaluated the biodegradability of poly-L-glutamic acid (PLGA) and PLL using the aforementioned system and demonstrated that PLGA and PLL were degraded and eluted in the low-MW fractions of gel filtration chromatography [21]. This finding was in good agreement with the results for unmodified PLL in the present study (Figure 4C). Ser-PLL is a polymeric bilayer with two different types of amino acids. Its surface Ser layer was released into the kidneys at 3 h after intravenous injection (Figure 4D). Gel filtration chromatography of unmodified PLL and Ser-PLL, together with the results of quick kidney distribution (Figure 1), indicated that both substances were digested in the kidneys within 3 h after their distribution. Ser-PAMAM was synthesized by coupling Ser with PAMAM via the same amido linkage as Ser-PLL. Nevertheless, these two conjugates differed in terms of the release rates of their surface Ser layers. Ser-PLL released its surface Ser layer faster than Ser-PAMAM (Figure 4B,D) possibly because the dendritic structure of Ser-PAMAM partially blocked peptidase access and, by extension, conjugate digestion in the kidneys. These discoveries, along with the fact that unmodified PAMAM is more slowly biodegradable than unmodified PLL (Figure 4A,C), indicate that Ser-PLL has superior biodegradability to dendritic Ser conjugates.

The ^90^Y isotope is a β-emitter with a high endpoint energy. It is classified as a first-order unique forbidden emitter and is expected to be suitable for radionuclide therapy against various tumor cells [12,13,14]. As ^90^Y beta rays do not penetrate from outside the body, however, it is impossible to confirm their tissue distribution by imaging. Therefore, ^111^In was selected for the tissue distribution analysis as it is a gamma-nucleus species with high penetrating power, imageable gamma rays, and a physical half-life near that of ^90^Y. It was confirmed from the results of the animal experiments that there was no significant difference between ^111^In and ^90^Y [12,13] in terms of their in vivo tissue distributions. Hence, both ^90^Y-labeled Ser-PLL and ^111^In-labeled Ser-PLL would be similar in this respect. In the mouse RCC model, cancer cells were inoculated into the proximal tubules and their vicinity in the renal cortex as these structures are associated with RCC pathogenesis. Here, specific kidney renal cortex distributions of ^111^In-labeled Ser-PLL were observed. According to the intrarenal distribution study, FITC-labeled Ser-PLL accumulated mainly in the proximal tubules. Thus, we postulate that ^90^Y-labeled Ser-PLL would also accumulate in the proximal tubules, where the tumor localizes. Furthermore, beta-rays suppressed kidney tumor growth.

It was reported that ^90^Y readily induces bone marrow depression (BMD) [14]. However, neither free ^90^Y nor ^90^Y-labeled Ser-PLL affected the white blood cell counts here. Therefore, the beta ray energy derived from ^90^Y fell below the BMD threshold. Moreover, the selective renal distribution of Ser-PLL also avoided inducing BMD.

It was previously reported that glomerular capillary endothelial and mesangial cells play important roles in the pathogenesis of radiation-induced nephropathy. In addition, ^90^Y caused radiation damage mostly in the glomeruli [32]. Based on the tissue and intrarenal distribution analyses performed here, Ser-PLL should pass through the glomeruli after intravenous injection. Hence, we suggest that the glomerular distribution and radiation nephrotoxicity of ^90^Y could be avoided by using Ser-PLL. Detailed toxicological studies on ^90^Y-Ser-PLL are required before it can be clinically applied. Notwithstanding, the results of this work indicated that ^90^Y-labeled Ser-PLL effectively suppresses renal tumor growth while attenuating ^90^Y nephrotoxicity. To the best of our knowledge, this is the first study to demonstrate suppression of renal tumor growth by kidney-targeting of ^90^Y after intravenous injection.

## 5. Conclusions

The present study demonstrated that Ser modification was effective in kidney-targeted drug delivery using a linear PLL as the drug carrier core. Ser-PLL selectively accumulated in the renal proximal tubules which are the sites of renal cell carcinoma (RCC) pathogenesis. Ser-PLL had higher biodegradability than the dendritic Ser conjugate. Renal tumor growth was effectively suppressed by ^90^Y-mediated kidney targeting via Ser-PLL. The foregoing findings indicate that Ser-PLL is promising as a biodegradable kidney-targeted drug carrier for radionuclide therapy against RCC.

## Figures and Tables

**Figure 1 pharmaceutics-14-01946-f001:**
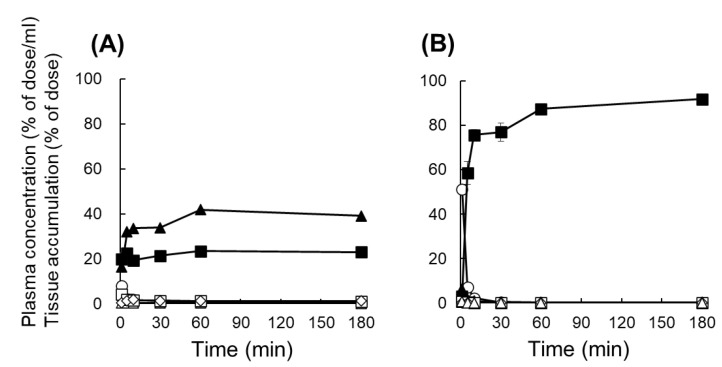
Plasma concentrations and tissue distributions of ^111^In-labeled poly-L-lysine (PLL) (**A**) and ^111^In-labeled Ser-poly-L-lysine (Ser-PLL) (**B**) after intravenous injection into mice. Data are means ± SE for three mice. ◯, plasma; ▲, liver; ■, kidney; ◇, spleen; △, heart; □, lung.

**Figure 2 pharmaceutics-14-01946-f002:**
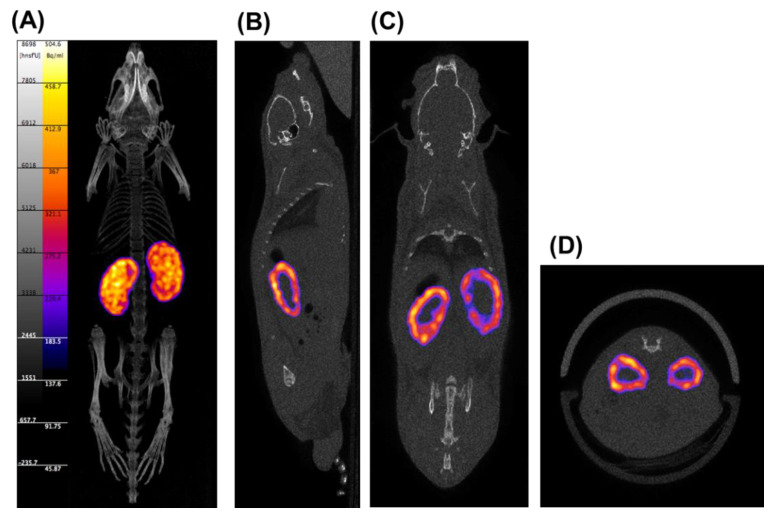
SPECT/CT imaging 180 min after intravenous injection of ^111^In-labeled Ser-poly-L-lysine (Ser-PLL) in mice. (**A**) 3D imaging; (**B**) sagittal plane; (**C**) coronal plane; (**D**) transverse plane.

**Figure 3 pharmaceutics-14-01946-f003:**
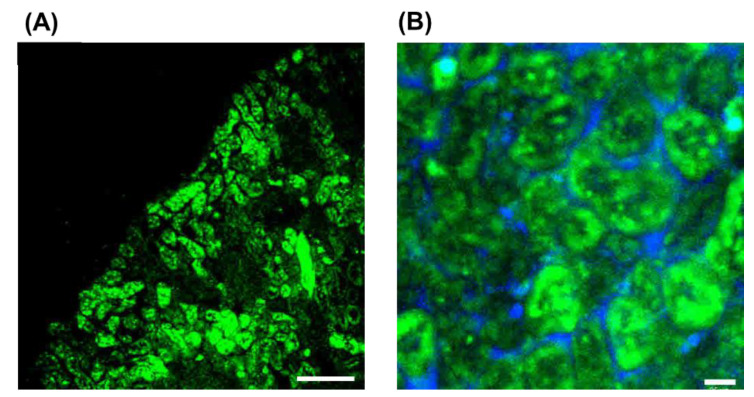
Intrarenal distribution of FITC-labeled Ser-poly-L-lysine (Ser-PLL) in tissue sections 60 min after intravenous injection into mice. (**A**) Renal cortex (scale bar = 200 μm) and (**B**) magnified image of cortex (scale bar = 25 μm). Fluorescence intensity was observed under a laser-scanning confocal microscope.

**Figure 4 pharmaceutics-14-01946-f004:**
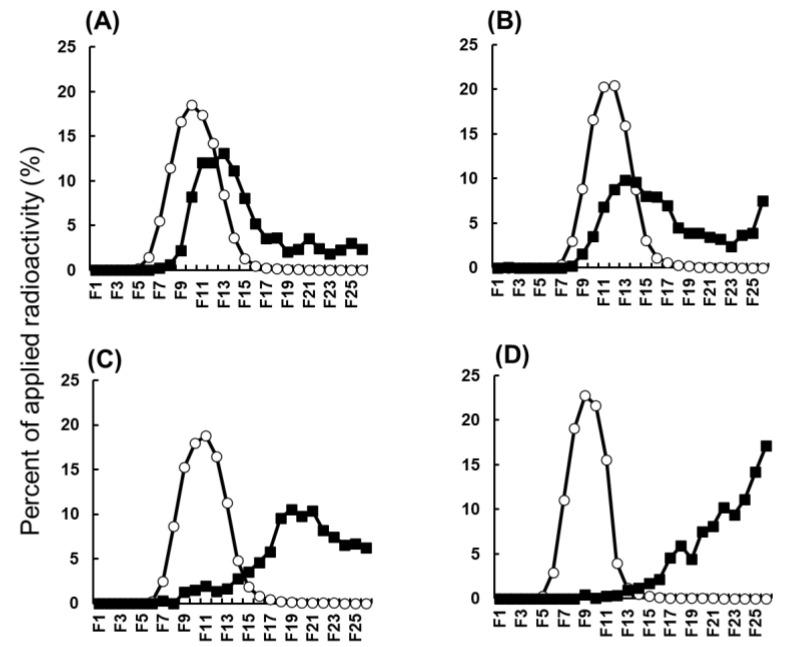
Gel filtration patterns of ^111^In-radioacticity in kidney homogenates from mice injected with ^111^In-labeled polyamidoamine (PAMAM) (**A**), ^111^In-labeled Ser-PAMAM (**B**), ^111^In-labeled poly-L-lysine (PLL) (**C**), and ^111^In-labeled Ser-poly-L-lysine (Ser-PLL) (**D**). ◯, pre-injection (intact compounds); ■, 180 min post-injection.

**Figure 5 pharmaceutics-14-01946-f005:**
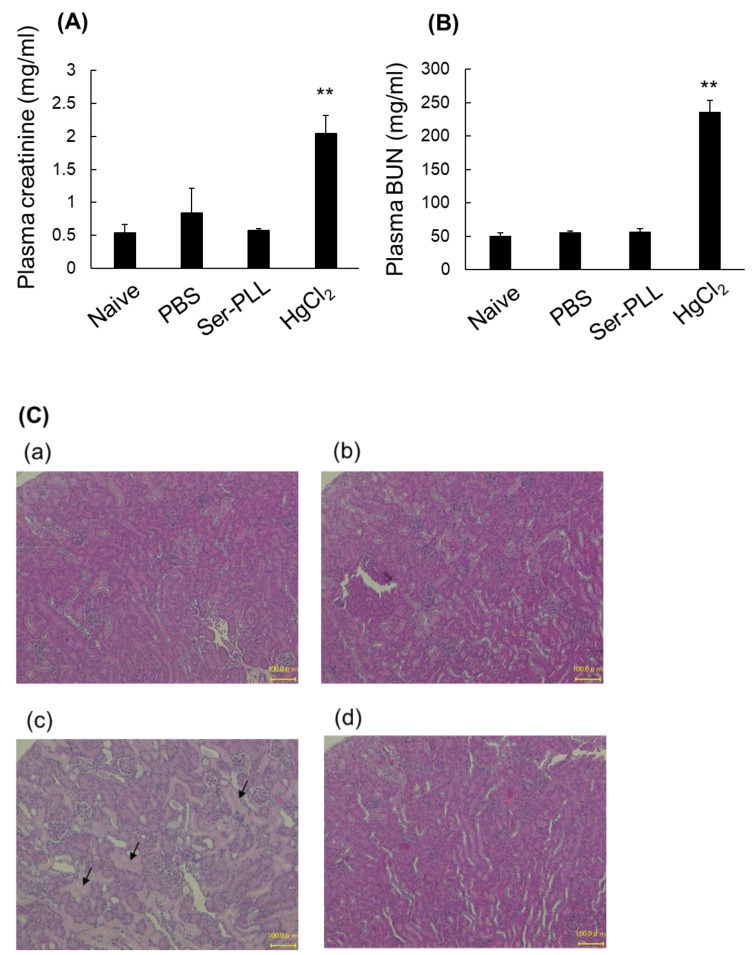
Plasma creatinine (**A**) and BUN (**B**) levels after continuous intravenous Ser-poly-L-lysine (Ser-PLL) injection for 5 d. Data are means ± SE for ≥ three mice. ** statistically significant difference compared with the naïve group (*p* < 0.01). (**C**) Histological examination of kidneys of mice in the naïve group (**a**), PBS (**b**), HgCl_2_ (**c**), and Ser-poly-L-lysine (Ser-PLL) (**d**). (scale bar = 100 μm).

**Figure 6 pharmaceutics-14-01946-f006:**
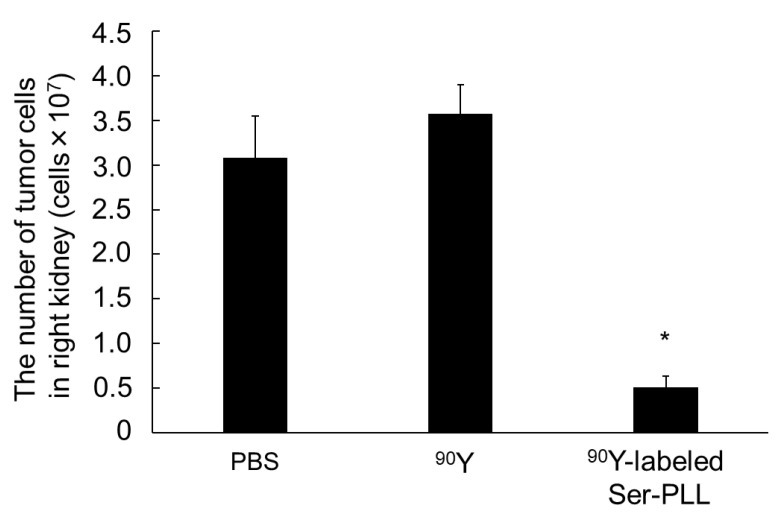
Effects of ^90^Y-labeled Ser-poly-L-lysine (Ser-PLL) on kidney tumor growth after tumor induction and intravenous injections. Number of Colon26/Luc cells in kidneys of mice intravenously injected either with free ^90^Y or with ^90^Y-labeled Ser-poly-L-lysine (Ser-PLL). Mice were sacrificed 14 d after tumor inoculation. Data are means ± SE for seven mice. * statistically significant difference compared with PBS group (*p* < 0.05).

**Figure 7 pharmaceutics-14-01946-f007:**
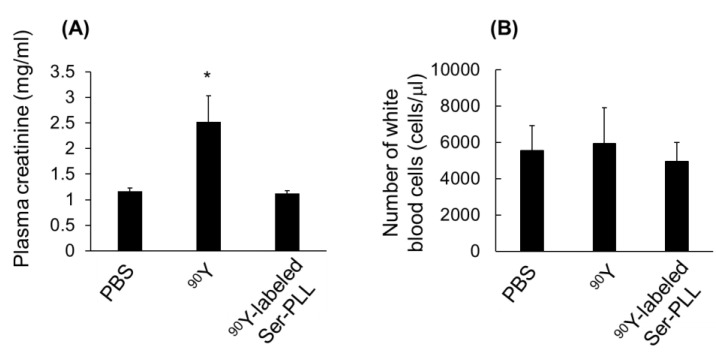
Plasma creatinine (**A**) and white blood cell counts (**B**) after intravenous injection of ^90^Y-labeled Ser-poly-L-lysine (Ser-PLL) at 0.3 MBq/mouse into mice inoculated with Colon26/Luc cells in right kidney cortex. (**C**) Histological examination of left kidneys of mice injected with ^90^Y-labeled Ser-poly-L-lysine (Ser-PLL) at 0.3 MBq/mouse and inoculated with Colon26/Luc cells in right kidney cortex. PBS (**a**), free ^90^Y (**b**), ^90^Y-labeled Ser-poly-L-lysine (Ser-PLL) (**c**). (scale bar = 100 μm) Mice were sacrificed 14 d after tumor induction. Data are means ± SE of ≥ three mice. * statistically significant difference compared with PBS group (*p* < 0.05).

**Table 1 pharmaceutics-14-01946-t001:** Physicochemical properties of poly-L-lysine and Ser-poly-L-lysine.

Compound	Diameter (nm)	Zeta Potential (mV)
Poly-L-lysine	3.3 ± 0.5	8.9 ± 1.6
Ser-poly-L-lysine	4.1 ± 0.9	6.6 ± 3.7

## Data Availability

Not applicable.

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
