# Peer review of "L-Serine-Modified Poly-L-Lysine as a Biodegradable Kidney-Targeted Drug Carrier for the Efficient Radionuclide Therapy of Renal Cell Carcinoma"

_pharmaceutics, 2022, doi:10.3390/pharmaceutics14091946_

Round 1

Reviewer 1 Report

The synthetic approach used in the research article entitled "L-serine-modified poly-L-lysine as a biodegradable kidney targeted drug carrier for the efficient radionuclide therapy of renal cell carcinoma" is very useful thought provoking. However, unfortunately, authors did not do justice with the idea of synthesis. I have few concerns and these questions seem intriguing to me that

1) The only benefit of using amino acids like serine and PLL is biocompatibility and biodegradabilty? If yes, then how can targeted outcomes in kidney cancer can be achieved via this combination

2) Exact targeting mechanism of action of Serine and Lysine is missing

3) Conjugation of serine onto lysine is mentioned without any confirmatory tests like NMR and FTIR. We cannot estimate success of conjugation without performing these tests.

4) ) Histological examination of kidneys of mice is poorly presented in Figures

Author Response

To Reviewer 1

First of all, we would like to express our appreciation to the reviewers for raising important issues and giving us helpful suggestions. We have revised our manuscript in the light of the reviewers’ comments. Our replies to each of the reviewer’s inquiries and the revised points are as follows.

The synthetic approach used in the research article entitled "L-serine-modified poly-L-lysine as a biodegradable kidney targeted drug carrier for the efficient radionuclide therapy of renal cell carcinoma" is very useful thought provoking. However, unfortunately, authors did not do justice with the idea of synthesis. I have few concerns and these questions seem intriguing to me that

1) The only benefit of using amino acids like serine and PLL is biocompatibility and biodegradabilty? If yes, then how can targeted outcomes in kidney cancer can be achieved via this combination

2) Exact targeting mechanism of action of Serine and Lysine is missing

A> We appreciate your thoughtful comment.

Effective delivery of drug carriers selectively to the kidney is challenging because of their uptake by the reticuloendothelial system in the liver and spleen. This limits effective treatment of kidney diseases and results in side effects. Recently, we found that after L-serine (Ser)-modified polyamideamine (PAMAM) dendrimer (Ser-PAMAM) specifically accumulated in the proximal tubules of the renal cortex, a pattern that is associated with the pathogenesis of renal cell carcinoma [8]. (Page 1, second paragraph)

In our previous study, we demonstrated that Ser-PAMAM was distributed in the proximal tubules via glomerular filtration and the hydroxyl and amino groups of Ser play important roles in kidney targeting. Micropinocytosis and caveola-mediated endocytosis have been implicated in the affinity of Ser-PAMAM for proximal tubules [8]. Therefore, we postulate that Ser-PLL is distributed in the proximal tubules in the same manner as Ser-PAMAM, due to the size of Ser-PLL (~5 nm), which is smaller than the glomerular filtration size cutoff (10 nm) [27-29]. Moreover, both Ser-PLL and Ser-PAMAM have the same surface functional groups. The foregoing results and those for renal Ser-PLL(H) accumulation (PLL MW range = 15,000–30,000) (Fig. S2B) suggest that Ser-PLL (PLL MW = 4,000–15,000) is the optimal size for kidney-targeted delivery mediated by glomerular filtration.

Although further studies are needed to elucidate the more detailed mechanism on the kidney-targeting using Ser-modification, the present study indicates that Ser-PLL is a promising drug carrier for the kidney-targeted drug delivery.

In addition to advantage of Ser-PLL on high biocompatibility and biodegradability, Ser-PLL had some advantages for kidney targeting ability compared with Ser-PAMAM (Fig. S2A). Ser-PLL had greater kidney accumulation than Ser-PAMAM (Fig. S2A), possibly because the estimated average number of conjugated Ser residues was 46 in Ser-PLL whereas Ser-PAMAM conjugated only 32. The number of conjugated Ser residues is proportional to renal affinity [8]. These results indicate that Ser modification could be applied in a kidney-targeted delivery system using linear macromolecular drug carriers in addition to branched macromolecular drug carriers.

Furthermore, we demonstrated that kidney tumor growth was significantly inhibited by administration of the beta-emitter 90Y combined with Ser-PLL in a murine RCC model (Fig. 6). To the best of our knowledge, this is the first study to demonstrate suppression of renal tumor growth by kidney-targeting of 90Y after intravenous injection.

In the revised manuscript, we newly added Figure S2A and following sentences in the text.

“In our previous study, we demonstrated that Ser-PAMAM was distributed in the proximal tubules via glomerular filtration. Micropinocytosis and caveola-mediated endocytosis have been implicated in the affinity of Ser-PAMAM for proximal tubules [8]. Therefore, we postulate that Ser-PLL is distributed in the proximal tubules in the same manner as Ser-PAMAM, due to the size of Ser-PLL (~5 nm), which is smaller than the glomerular filtration size cutoff (10 nm) [27-29]. Moreover, both Ser-PLL and Ser-PAMAM have the same surface functional groups. The foregoing results and those for renal Ser-PLL(H) accumulation (PLL MW range = 15,000–30,000) (Fig. S2B) suggest that Ser-PLL (PLL MW = 4,000–15,000) is the optimal size for kidney-targeted delivery mediated by glomerular filtration.” (Page 10, second paragraph)

“Ser-PLL had slightly greater kidney accumulation than Ser-PAMAM (Fig. S2A), possibly because the estimated average number of conjugated Ser residues was 45 in Ser-PLL whereas Ser-PAMAM conjugated only 32. The number of conjugated Ser residues is proportional to renal affinity [8]. These results indicate that Ser modification could be applied in a kidney-targeted delivery system using linear macromolecular drug carriers in addition to branched macromolecular drug carriers.” (Page 9, first paragraph)

“To the best of our knowledge, this is the first study to demonstrate suppression of renal tumor growth by kidney-targeting of 90Y after intravenous injection.” (Page 11, first paragraph)

We hope our explanation and revisions address your comments.

3) Conjugation of serine onto lysine is mentioned without any confirmatory tests like NMR and FTIR. We cannot estimate success of conjugation without performing these tests.

A> We appreciate your thoughtful comment.

In the present study, the obtain Ser-PLL was identified by 1H nuclear magnetic resonance (NMR) spectroscopy (AM-300 FT-NMR spectrometer; Bruker Corp., Billerica, MA, USA) in deuterated water (D2O). In the PLL 1H NMR spectrum, peaks were observed at d 4.18 (s, a–CH), 2.81 (m, e–CH2), 1.82–1.05 (s, b–CH2, g–CH2, and d–CH2). In the Ser-PLL 1H NMR spectrum, the peaks corresponding to Ser appeared at δ 3.72–3.85 (m, CH2), and the integral ratio of Ser methylene protons to PLL methylene protons indicated that the desired product was obtained (the degree of modification of the Ser moiety linked to the amino groups of PLL was 98.8%) (Figure S1). As the polymerization degree of PLL is estimated to be approximately 19-72 units with a MW range of 4,000–15,000, these results indicate that the average composition of Ser units of the obtained Ser-PLL was 45, with an approximate Ser-PLL molar mass of 10 kDa.

In the revised manuscript, we added the following sentences in the text.

“In the PLL 1H NMR spectrum, peaks were observed at d 4.18 (s, a–CH), 2.81 (m, e–CH2), 1.82–1.05 (s, b–CH2, g–CH2, and d–CH2). In the Ser-PLL 1H NMR spectrum, the peaks corresponding to Ser appeared at δ 3.72–3.85 (m, CH2), and the integral ratio of Ser methylene protons to PLL methylene protons indicated that the desired product was obtained (the degree of modification of the Ser moiety linked to the amino groups of PLL was 98.8%) (Figure S1). As the polymerization degree of PLL is estimated to be approximately 19-72 units with a MW range of 4,000–15,000, these results indicate that the average composition of Ser units of the obtained Ser-PLL was 45, with an approximate Ser-PLL molar mass of 10 kDa.” (Page 3, first paragraph)

4) ) Histological examination of kidneys of mice is poorly presented in Figures

A> We appreciate your thoughtful comment.

In the revised manuscript, we added the arrows where damage and necrosis were occurred in the Figures 5C and 7C. In addition, we replaced Figure 5C to the pictures with wide view to cover the wide range.

The underline parts were also added in the text as follows.

“The kidney sections disclosed severe gap junction damage and necrosis (arrows) after the HgCl2 treatment. However, Ser-PLL had negligible adverse effects on renal structure, and the tissue from Ser-PLL-treated mice was similar to that from PBS-treated and naïve mice (Fig. 5C).” (Page 7, first paragraph)

“Partial glomerular damage and necrosis (arrow) was observed in the kidney sections of the free 90Y group.” (Page 8, first paragraph)

Author Response

To Reviewer 2

First of all, we would like to express our appreciation to the reviewers for raising important issues and giving us helpful suggestions. We have revised our manuscript in the light of the reviewers’ comments. Our replies to each of the reviewer’s inquiries and the revised points are as follows.

The paper described Ser-PLL polymers carrying radionuclide In/Y for renal cell carcinoma treatment/imaging. The strong targeting ability of the polymer toward kidney tissues was proved, the biodegradability, biodistribution and the effectiveness of anti-tumor activity over animal model were shown. The potential of this carrier for delivery therapeutic agents is sufficiently demonstrated. While the scientific merits of this article are high, there are rooms for improvement as listed below:

  1. The polymer’s molecular weight distribution and its composition are important items for the physicochemical properties for this carrier.

An NMR spectrum is needed.

A> We appreciate your thoughtful comment.

In the present study, the obtain Ser-PLL was identified by 1H nuclear magnetic resonance (NMR) spectroscopy (AM-300 FT-NMR spectrometer; Bruker Corp., Billerica, MA, USA) in deuterated water (D2O) (Figure S1). In the PLL 1H NMR spectrum, peaks were observed at d 4.18 (s, a–CH), 2.81 (m, e–CH2), 1.82–1.05 (s, b–CH2, g–CH2, and d–CH2). In the Ser-PLL 1H NMR spectrum, the peaks corresponding to Ser appeared at δ 3.72–3.85 (m, CH2), and the integral ratio of Ser methylene protons to PLL methylene protons indicated that the desired product was obtained (the degree of modification of the Ser moiety linked to the amino groups of PLL was 98.8%) (Figure S1). As the polymerization degree of PLL is estimated to be approximately 19-72 units with a MW range of 4,000–15,000, these results indicate that the average composition of Ser units of the obtained Ser-PLL was 45, with an approximate Ser-PLL molar mass of 10 kDa.

In the revised manuscript, we added the following sentences in the text.

“In the Ser-PLL 1H NMR spectrum, the peaks corresponding to Ser appeared at δ 3.72–3.85 (m, CH2), and the integral ratio of Ser methylene protons to PLL methylene protons indicated that the desired product was obtained (the degree of modification of the Ser moiety linked to the amino groups of PLL was 98.8%) (Figure S1). As the polymerization degree of PLL is estimated to be approximately 19-72 units with a MW range of 4,000–15,000, these results indicate that the average composition of Ser units of the obtained Ser-PLL was 45, with an approximate Ser-PLL molar mass of 10 kDa.” (Page 3, first paragraph)

  1. In biodegradability data shown in Fig.4, the volume of each fraction needs to be specified. Also, it is quite surprising that the polymer should have been degrading so fast in 180 min. Independent in vitro degradation tests are desirable to validate the unexpected results.

A> We appreciate your thoughtful comment.

Following the reviewer’s comments, we added the volume of each fraction in the text.

In the revised manuscript, we added the following sentences in the text.

“The radioactivity levels of each fraction (10 drops, 0.35 ml) were measured.” (Page 4, second paragraph)

Post-elution 111In radioactivity detection by gel filtration chromatography was established as a method to assess the biodegradability of macromolecular drug carriers [19].

In our preliminary data, we confirmed 111In-labeled Ser-PLL was stable (>90%) for 3 h after incubation of 111In-labeled Ser-PLL in plasma at 37°C, which was evaluated by gel filtration chromatography.

Furthermore, in the tissue distribution study, we demonstrated that 111In-labeled Ser-PLL rapidly eliminated from the blood circulation and approximately 75 % of the dose quickly accumulated in the kidney in 10 min (Figure 1).

These results indicate that most Ser-PLL quickly eliminated from the blood circulation and distributed to the kidney within 10-60 min and Ser-PLL was rapidly biodegraded in the kidney after the distribution.

“In the revised manuscript, we changed the following sentences in the text.

“Gel filtration chromatography of unmodified PLL and Ser-PLL indicated that both substances are digested in the kidney within 3 h after intravenous injection.” (Page 10, third paragraph)

was changed to

“Gel filtration chromatography of unmodified PLL and Ser-PLL, together with the results of quick tissue distribution (Fig. 1), indicated that both substances were digested in the kidneys within 3 h after their distribution.” (Page 10, third paragraph)

We hope our explanation and revisions address your comments.

  1. The suitability of the RCC animal model needs to be established. Are there relevant references that can support this claim?

A> We appreciate your thoughtful comment. In the present study, we established RCC animal model using a previously published method with slight modifications [20]. Following the reviewer’s comments, we cited the following relevant reference in the text.

[20] Norian LA, Kresowik TP, Rosevear HM, James BR, Rosean TR, Lightfoot AJ, Kucaba TA, Schwarz C, Weydert CJ, Henry MD, Griffith TS. Eradication of metastatic renal cell carcinoma after adenovirus-encoded TNF-related apoptosis-inducing ligand (TRAIL)/CpG immunotherapy. PLoS One. 2012;7(2):e31085. doi: 10.1371/journal.pone.0031085. Epub 2012 Feb 1. PMID: 22312440; PMCID: PMC3270031.

In the revised manuscript, we added the following sentence in the text.

“The RCC model was established using a previously published method with slight modifications [20]. (Page 4, fourth paragraph)

Reviewer 3 Report

This manuscript covered the radionuclide therapy of kidney cancer using biodegradable kidney targeted polymers. It showed the proof-of-concepts through in vitro and in vivo studies. Thus, it is publishable in this journal because it did provide new insight, and the depth of manuscript is well-organized and mature because all of contents are new and innovative. Therefore, it is enough to publish in this journal after minor revision.

1.     The author mentioned that “drug carrier” but Ser-PLL seems like a drug conjugate. Thus, it needs to be fixed.  

2.     The authors need to show the release profile of Ser from Ser-PLL. It will prove the stability of Ser-PLL within the whole experimental time frame.

Author Response

To Reviewer 3

First of all, we would like to express our appreciation to the reviewers for raising important issues and giving us helpful suggestions. We have revised our manuscript in the light of the reviewers’ comments. Our replies to each of the reviewer’s inquiries and the revised points are as follows.

This manuscript covered the radionuclide therapy of kidney cancer using biodegradable kidney targeted polymers. It showed the proof-of-concepts through in vitro and in vivo studies. Thus, it is publishable in this journal because it did provide new insight, and the depth of manuscript is well-organized and mature because all of contents are new and innovative. Therefore, it is enough to publish in this journal after minor revision.

  1. The author mentioned that “drug carrier” but Ser-PLL seems like a drug conjugate. Thus, it needs to be fixed.  

A> We appreciate your thoughtful comment. We agree that Ser-PLL is a conjugate type of drug carrier. However, poly(amino acid) is generally categorized as macromolecular drug carrier for conjugation [24][25]. Therefore, we left “drug carrier” in the text.

  1. The authors need to show the release profile of Ser from Ser-PLL. It will prove the stability of Ser-PLL within the whole experimental time frame.

A> We appreciate your thoughtful comment.

In our preliminary data, we confirmed 111In-labeled Ser-PLL was stable for 3 h after incubation of 111In-labeled Ser-PLL in plasma at 37°C, which was evaluated by gel filtration chromatography.

Furthermore, we demonstrated that 111In-labeled Ser-PLL rapidly eliminated from the blood circulation and approximately 75% of the dose quickly accumulated in the kidney in 10 min (Figure 1).

These results indicate that most Ser-PLL quickly eliminated from the blood circulation and distributed to the kidney within 10-60 min and Ser-PLL was rapidly biodegraded in the kidney after the distribution.

We agree that the release profile of Ser from Ser-PLL is important to accurately simulate the stability of Ser-PLL in vivo. We are now considering the idea of investigating this point in a future paper.

“In the revised manuscript, we added the following sentences in the text.

“Gel filtration chromatography of unmodified PLL and Ser-PLL indicated that both substances are digested in the kidney within 3 h after intravenous injection.” (Page 10, third paragraph)

was changed to

“Gel filtration chromatography of unmodified PLL and Ser-PLL, together with the results of quick tissue distribution (Fig. 1), indicated that both substances were digested in the kidneys within 3 h after their distribution.” (Page 10, third paragraph)

We hope our explanation and revisions address your comments.
